# Molecular Aspects in the Development of Type 2 Diabetes and Possible Preventive and Complementary Therapies

**DOI:** 10.3390/ijms25169113

**Published:** 2024-08-22

**Authors:** Laura Simon-Szabó, Beáta Lizák, Gábor Sturm, Anikó Somogyi, István Takács, Zsuzsanna Németh

**Affiliations:** 1Department of Molecular Biology, Semmelweis University, Tuzolto u. 37-47, 1094 Budapest, Hungary; szabo.laura@semmelweis.hu (L.S.-S.); lizak.beata@semmelweis.hu (B.L.); 2Directorate of Information Technology Basic Infrastructure and Advanced Applications, Semmelweis University, Üllői út 78/b, 1082 Budapest, Hungary; sturm.gabor@semmelweis.hu; 3Department of Internal Medicine and Hematology, Semmelweis University, Baross u., 1085 Budapest, Hungary; somogyi.aniko@med.semmelweis-univ.hu; 4Department of Internal Medicine and Oncology, Semmelweis University, Koranyi S. u 2/a, 1083 Budapest, Hungary; takacs.istvan@semmelweis.hu

**Keywords:** insulin signaling, T2DM, insulin resistance, glucose intolerance, life style, prevention, complementary therapy, conventional therapy

## Abstract

The incidence of diabetes, including type 2 diabetes (T2DM), is increasing sharply worldwide. To reverse this, more effective approaches in prevention and treatment are needed. In our review, we sought to summarize normal insulin action and the pathways that primarily influence the development of T2DM. Normal insulin action involves mitogenic and metabolic pathways, as both are important in normal metabolic processes, regeneration, etc. However, through excess energy, both can be hyperactive or attenuated/inactive leading to disturbances in the cellular and systemic regulation with the consequence of cellular stress and systemic inflammation. In this review, we detailed the beneficial molecular changes caused by some important components of nutrition and by exercise, which act in the same molecular targets as the developed drugs, and can revert the damaged pathways. Moreover, these induce entire networks of regulatory mechanisms and proteins to restore unbalanced homeostasis, proving their effectiveness as preventive and complementary therapies. These are the main steps for success in prevention and treatment of developed diseases to rid the body of excess energy, both from stored fats and from overnutrition, while facilitating fat burning with adequate, regular exercise in healthy people, and together with necessary drug treatment as required in patients with insulin resistance and T2DM.

## 1. Introduction

### 1.1. History of Diabetes

The symptoms of diabetes mellitus have been know from ancient times in medical history. Descriptions of the characteristic symptoms such as extreme thirst, excessive drinking, and large amount of sweet urine were mentioned by physicians from ancient Egypt, India, and China [1,2,3,4]. In the Middle Ages, complications were also attributed to the disease like skin infections, gangrene, and eyesight problems. The first detailed information of the disease symptoms and progression coming from Aretaeus of Cappadocia with the term *diabetes* in the 2nd century. The term *mellitus* was included by Thomas Willis in the 17th century [5].

Paul Langerhans first identified those tissue clamps in 1869, recently called “*Langerhans islands*”, which are located in the pancreatic tissue and their secreted solutions have key roles in metabolic processes [6].

The connection of the pancreas, as an organ, with diabetes was discovered in 1889 by von Mering and Minkowski, who detected the presence of sugar in dog urine after pancreatectomy; however, they probably did not know about the existence of Langerhans islands [1,7]. The first successful extraction of “*isletin*”, the former name of insulin, was performed from a dog by Banting and Best in 1921 and later that year from an adult cow. The first injection of insulin to a 14-year-old diabetic patient was applied in 1920, and this treatment finally confirmed the causality link between diabetes and the insulin secretion product of the pancreas [8].

### 1.2. Type 2 Diabetes Mellitus

Diabetes is a metabolic disease that leads to hyperglycemia caused by two main factors: defective insulin secretion and impaired insulin action or both [9]. Insulin is produced by the β-cells of Langerhans islands in the pancreas. This molecule, consisting of 51 amino acids, is not only a hormone that regulates the uptake and storage of glucose from the blood but is also an important surviving signal of the cells. Glucagon, another product of the Langerhans islands, is released by the α-cells and works in an opposite way to insulin, stimulating the conversion of glycogen to glucose, and increasing the glucose level in blood [10]. Diabetes is subdivided into insulin-dependent (type 1, T1DM) and non-insulin-dependent (type 2, T2DM) clinical forms [11]. In both forms, besides hyperglycaemia, there are other symptoms indicating the presence of the disease, i.e., weight loss in non-obese patients, excessive urine production, compensatory thirst, blurred vision, and lethargy [10]. T1DM is caused by autoimmune deterioration of pancreatic b-cells, which completely abolishes insulin secretion [9]. However, in T2DM, functional defects of b-cells occur together with impaired insulin effect on target cells, i.e., insulin resistance (IR) [12].

Approximately 90% of all diabetic cases belong to the T2DM form. T2DM became one of the leading chronic diseases causing death worldwide [12,13,14]. According to health surveys and predictions, the prevalence of diabetes will affect approximately 700 million people by 2045 [15,16]. The role of genetic factors in the pathomechanism of T2DM is unquestionable, but still not completely understood. Sedentary lifestyle, the lack of physical activity, and excess calorie intake are known as the main factors in the development of this disease [17]. Since this is partly a lifestyle-related disease, it can be prevented, but this requires extensive knowledge about the development of the disease. In this review, we aimed to highlight those molecular aspects with the main focus on IR, which can be linked to lifestyle-related metabolic changes and, thus, can shed light on possible prevention strategies.

## 2. Metabolic Syndrome (MetS)

One of the key factors in the development of type 2 diabetes is the phenomenon of metabolic syndrome (MetS) [18,19,20]. Initially, it was called “syndrome X” by Reaven [21]. The name MetS came later and was characterized by four main factors: hypertension, dyslipidemia, abdominal obesity, and IR [22,23], in addition to others such as high fasting glucose level, increased risk of blood clotting, and a tendency to develop inflammation [20]. MetS is a complex metabolic disorder with a background of a low-level systemic inflammation [19,20,23].

### 2.1. Hypertension

Hypertension is diagnosed when the blood pressure is higher than 140/90 mmHg [24,25]. This can develop from both by hereditary genetic background and by environmental factors such as smoking, excess body weight, high sodium and/or low potassium intake, physical inactivity, or high level of alcohol consumption, which all can increase the risk of elevated blood pressure [26,27]. Hypertension is more frequent in patients with T2DM than in the healthy population [28]. It can be assumed that microcirculatory impairment, resulting in high blood pressure, may have several causes. In patients with T2DM, the elevated glucose level can cause arterial stiffness and thickness and damage in the endothelial function [29]. Chronic hyperglycaemia is increasing the level of glycation end-products (AGEs), which can lead to increased reactive oxygen species (ROS) damaging the endothelial layer, but also can decrease endothelial function by inhibiting the nitrogen–monoxide (NO) production [30,31]. As they affect the microvasculature, both a hypertensive state and the presence of T2DM are increased risk factors for cardiovascular diseases (CVD) [28].

### 2.2. Dyslipidemia

Dyslipidemia is a quantitative change in triglycerides (TGs) and/or total cholesterol in the blood [32]. Another definition called atherogenic dyslipidemia is used more typically in patients with abdominal obesity, described by moderate elevation of TGs, very-low-density lipoprotein (VLDL) Apo B, and non-high-density lipoprotein cholesterol (non-HDL-C) levels of the plasma [33]. Diabetic dyslipidemia is similarly defined by elevated level of TGs, low-density lipoprotein cholesterol (LDL-C), and small dense LDL particles, and, additionally, with decreased levels of HDL-C [22,34]. Dyslipidemia is a consequence of IR, which is the result of an increased influx of enhanced level free fatty acid (FFA) to the liver due to glucose intolerance and compensatory elevated insulin levels [21,35,36]. Although under physiological conditions insulin inhibits the secretion of triglyceride-rich VLDLs into the circulation, in the IR state, the increased influx of FFA into the liver increases the synthesis of Apo-B-containing hepatic TG and triglyceride-rich VLDLs. In addition, increased TG promotes the exchange of cholesterol to TG in HDL-C and LDL-C, converting them into small and dense particles. These alterations increases the clearance of HDL—the reverse transporter of cholesterol (from the periphery to liver)—from the circulation, and the susceptibility of LDL to oxidation, resulting in its enhanced uptake by macrophages [36].

### 2.3. Abdominal Obesity

Dyslipidemia frequently develops in obese patients, and its atherogenic profile mainly occurs in the abdominal/visceral type of obesity [33]. Obesity affected approximately every third school-aged child and one in four adolescents and, additionally, around 60% of the adult population was overweight or obese in the WHO Europe Region in 2022 [37]. Although there are several ways to measure obesity, body mass index (BMI) is the most easily accessible, as it only requires the people’s weight and height, but it does not provide a direct measure of adiposity and location of fat tissues [38]. Overweight is defined when BMI is ≥25 kg/m^2^, while obesity is when it is ≥30 kg/m^2^ according to the WHO classification [37,39]. BMI is positively associated with both the total body fat mass and visceral obesity, as well as waist circumference; however, the last two represent a higher risk for CVD [40,41,42]. Moreover, a population-based study applying Mendelian randomization found that every singular standard deviation increment in the BMI increases the likelihood of T2DM by 67% and coronary artery disease by 20% [43].

### 2.4. Insulin Action and Insulin Resistance

#### 2.4.1. Normal Function of Insulin

Besides its hormonal function mediating glucose homeostasis, insulin is also an important survivor factor maintaining cell growth and proliferation [44]. Although insulin has acquired a primary function as anabolic, metabolic regulatory hormone in vertebrates [45], certain conditions, where insulin levels are elevated, can trigger mitogenic pathways through its receptor as well [46]. All the effects of insulin are mediated by its receptor, and depend on tissue targets, other hormones, biogenic amines, and their temporal relationship [47]. It is important to note that insulin-like growth factor 1 and 2 (IGF-1 and IGF-2) have similar mitogenic growth functions in vertebrates and both are the products of insulin gene family in mammals, similar to insulin. Since the receptors of these three molecules are very similar, they can activate many common downstream effectors by binding to their own receptors—moreover, all of them can bind to the insulin receptor as well [44].

#### 2.4.2. Insulin Pathway

The insulin receptor is a hetero-tetramer (didimer) glycoprotein. It has two extracellular regulatory ligand-binding alpha subunits and two transmembrane catalytic beta subunits with tyrosine (Y) kinase activity, all bounded by disulphide bondages [48,49]. When insulin binds to the alpha subunit of its receptor, it no longer inhibits the beta subunits, thus activating downstream signaling [50,51]. Interestingly, not only one but both the two distinct insulin-binding alpha sites can create complex with two insulin molecules [52], resulting in a T-shape, 4-insulin-saturated insulin receptor, as confirmed by cryo-electron microscopy [53,54]. This provided new insights into ligand specificity and selectivity, as a single receptor can bind to several different ligands, in addition to insulin, with multiple responses [53].

After insulin binds to its receptor, the insulin receptor substrate (IRS) molecules are attracted to the receptor through receptor autophosphorylation (Figure 1). Two types of IRS, namely IRS-1 and IRS-2, have so far been shown to be important in the transmission of insulin action [55,56]. These molecules are phosphorylated on their tyrosine residues by the autophosphorylated insulin receptor, the activation of which then mediates diverse downstream signaling events through either the mitogenic or metabolic pathways as shown in Figure 1 [44,57].

##### Mitogenic Pathway

The first component of the mitogenic pathway after IRS-1 is the growth factor receptor-bound protein 2 (GRB2) (Figure 1A). This is an adaptor protein with SH2 and prolin-rich SH3 domains, which can bind to phosphotyrosine residues, among others to IRS-1/IRS-2, with high affinity [58]. The subsequent molecule in the cascade is the son of sevenless (SOS), a guanine exchange factor (GEF), thus by catalyzing the exchange of GDP to GTP, regulates Ras protein at the plasma membrane [59]. This activates a mitogen-activated protein kinase (MAPK) cascade—RAF, MEK, ERK1/2—where, at the end, transcription factors such as c-jun, c-fos, and c-myc will be activated and induce cell proliferation (Figure 1A) [60].

##### Metabolic Pathway

When IRS-1 is recruited and activated at the insulin receptor, its tyrosine residues bind to the 85 kDa regulatory domain of phosphatidylinositol-3-kinase (PI3K) (Figure 1B). Thus, the catalytic subdomain of PI3K is activated and translocates to the plasma membrane, where it phosphorylates phosphatidyl-inositol-(4,5)-diphosphate (PI(4,5)P2) to phosphatidyl-inositol-(3,4,5)-triphosphate (PI(3,4,5)P3), in addition to proteins that can recognize PIP3, such as phosphatidyl-inositol dependent kinase 1 (PDK-1), protein kinase B (PKB) also known as Akt, as well as protein kinase C (PKC) isomers (PKCλ and PKCζ) anchor to the plasma membrane [10].

When Akt is activated by PDK-1, it dissociates from the plasma membrane, and initiate the activation of other proteins in the cytosol and in the nucleus [61]. There are several substrates of Akt, as detailed below, which have central roles in the development in T2DM.

The glycogen synthase kinase 3 (GSK3) is one of the Akt substrates in the cytosol. This is responsible for the regulation of glycogen synthase (GS), and the eukaryotic initiation factor 2B (eIF2B), among others. Inactivation of GSK3 by Akt activates GS, which then induces glycogenesis [62], as well as the eIF2B, which then enhances the cell growth (Figure 1B) [63].Forkhead transcription factors (FOXOs) are another targets of Akt. FOXOs acting in glucose metabolism increase the gene expression of key enzymes of gluconeogenesis, such as glucose-6-phosphatase and phospho-enol-pyruvate carboxykinase (PEPCK) [64]. When FOXOs are inactivated through phosphorylation by Akt, they exit the nucleus and are degraded by proteasomes; thereby Akt inhibit gluconeogenesis (Figure 1B) [65]. FOXOs also control gene expression that regulates the cell cycle, apoptosis, oxidative stress resistance, differentiation, and muscle atrophy as well as energy homeostasis. Thus, this can induce apoptosis through activation of FasL and Bim, and can promote cell-cycle arrest and stress resistance by induction of catalase and manganese superoxide dismutase (MnSOD) for inactivation of reactive oxygen species (ROS) as well as by facilitating DNA repair [66].The GTPase-activating protein Akt substrate of 160 kDa (AS160), also known as TBC1D4, is also an Akt substrate [67]. It is attached to vesicles of glucose transporter 4 (GLUT4)—an insulin-dependent transporter in muscle and adipose tissues [68]—and inhibits the efflux, and translocation of these vesicles from the Golgi to the plasma membrane through inactivation of Rab protein, thus restraining GLUT4 exocytosis (Figure 1B) [69]. When AS160 activity is blocked by Akt via phosphorylation, AS160 detaches from the GLUT4 vesicles and increases their efflux and exocytosis (Figure 1B) [70]. However, insulin-stimulated phosphorylation of AS160 impaired the skeletal muscle in T2DM [23]. Similarly to AS160, the AS160 paralog TBC1D1, which is highly expressed in skeletal muscle, is also phosphorylated by insulin, as well as by exercise and AMP kinase (AMPK); moreover, AMPK is suggested as the most robust regulator of this signaling molecule [23,71,72]. GLUT4 exocytosis can be also stimulated by PDK1 through PKCλ phosphorylation. PKCλ then stimulates Rab4 activity, which initiates GLUT4 exocytosis [73]. Additionally, TC10 lipid raft microdomain is also associated with and promotes GLUT4 fusion to the plasma membrane lipid rafts [74].Akt also can promote Ras-related C3 botulinum toxin substrate 1 (RAC1), a member of the Rho family of small GTPases, which enhance GLUT4 translocation by the regulation of cytoskeletal actin reorganization (Figure 1B) [75,76].Akt activates mammalian target of rapamycin complex 1 (mTORC1) by inhibiting tuberous sclerosis complex 2 (TSC2), which forms heterodimer with tuberous sclerosis complex 1 (TSC1) (Figure 1B) [77]. TSC2 is a GTPase-activating protein (GAP) for Ras homologue enriched in brain (Rheb) protein. When inhibition of Rheb is blocked by inactivated TSC complex, mTORC1 become activated, and then throughout the activation of p70 ribosomal protein S6 kinase (S6K1) and inhibition of eukaryotic translation initiation factor 4E-binding protein 1 (4E-BP1), the protein synthesis is enhanced (Figure 1B) [78].The pro-apoptotic proteins such as BCL2-antagonist of cell death (Bad) and Caspase-9 are phosphorylated, and inactivated by Akt as well (Figure 1B) [79,80].Akt, mTOR, and PKC λ/ζ can activate sterol regulatory binding proteins (SREBPs) to enhance lipid synthesis and regulate cholesterol homeostasis (Figure 1B). SREBPs are localized in the ER membrane and bind to the sterol cleavage protein (SCA) and sterol regulatory element (SRE). At low sterol levels, this complex is transported to the Golgi, where the transcription factor is deliberated by proteases. After translocation to the nucleus, genes of lipid synthesis will be induced (Figure 1B) [81,82].

#### 2.4.3. Insulin Resistance

In clinical practice, IR is diagnosed based on the results of an oral glucose tolerance test (OGTT), when blood glucose concentration fails to return to normal range after glucose consumption [83,84]. Physiologically, it is defined when glucose-consuming tissues are unable to respond to normal level of insulin, thus a higher amount is produced to maintain its normal function [76]. There are several measures of IR. The fasting glucose level, homeostasis model assessment of IR (HOMA-IR) [85], leptin/adiponectin ratio [86,87,88,89], and TG accumulation in skeletal muscle [90] are the best known, among others. However, the measurement of IR and the determination of its normal range, as well as the interpretation of the results, often cause difficulties in practice.

The underlying molecular mechanism of the development of IR is still under debate. The problem seems very complex, involving several changes in insulin signaling pathway as well as the energy metabolism of its target organs. Intramuscular TG accumulation is the most recognized, consistent marker of whole-body IR, which is associated with decreased use of fatty acid as energy source in skeletal muscle [90]. The highest FFA and insulin-dependent glucose consumption is conducted in skeletal muscle, which may explain its key role in whole-body IR as well as the widespread opinion that any defects in the utilization of these energy sources are the primary cause of the development of IR [91,92]. Another study suggested that defects mainly at the proximal level of insulin pathway (i.e., at insulin receptor, IRS-1, PI3K, and Akt) are the most likely reasons of IR in skeletal muscle [76,93,94], while others refer to a failure in GLUT4 translocation and/or exocytosis as the main reason [70].

IR frequently goes along with hypertension [95]. The molecular basis of this from one aspect is the missing function of insulin on vascular dilatation through NO production [96], while the sodium reabsorption in the kidney is retained [36]. Additionally, FFA can also mediate vasoconstriction, which is frequently elevated in IR as a result of increased body fat mass or obesity [97]. Moreover, the function of the ATP-dependent and Ca^2+^-activated K^+^ channels (K (ATP) and K (Ca^2+^)) channels are also attenuated in IR, which leads to vasoconstriction as well [98].

In the early phase of IR, increased insulin secretion is observed in beta cells to compensate for the insufficient insulin action [23,99]. It happens due to excessive calorie intake, which is regularly combined with a sedentary lifestyle and leads to obesity. In these situations, glucose cannot be taken-up properly by the targeted tissues following a meal, which leads to a state called glucose intolerance [23,100]. The human body, to manage the impaired insulin action, further enhances a compensatory insulin secretion, resulting in hyperinsulinemia. High insulin level is associated with elevated FFA in the circulation. The increased amount of glucose and FFA entering the cells may cause glucotoxicity or lipotoxicity [94], and this can lead to endoplasmic reticulum (ER) stress [93]. Increased cellular FFA during lipotoxicity disrupts ER–Golgi protein trafficking, causing protein overload in the ER, leading to ER stress, and increases levels of ER stress markers, including the ER chaperone BiP and the apoptosis inducer CHOP, and increases the phosphorylation of IRS -1 at Ser307, which results in its increased degradation [93,101,102,103]. FFA itself can trigger inflammation as well [104]. Moreover, increased visceral adipose tissues similarly induce inflammation by the production of inflammatory cytokines [105,106]. Eventually, all these processes mentioned above lead to beta cell exhaustion [107], with a final result of IR with decreased insulin signal activation [93,94] and/or subsequent blunted GLUT4 translocation and exocytosis as well [70]. In the worst case, this process ends with apoptotic cell death of the beta cells [107]. The progression of this metabolic condition can only be reversed, if at all possible, with a multi-level treatment approach including weight loss, healthy diet, regular physical activity, and, if needed, vitamin D supplementation, as well as appropriate drug treatment, e.g., metformin, among others, which reduces IR. Without any intervention, this process progresses to T2DM, where beta cells cannot compensate for the missing insulin action any more [20].

In the next sections, we summarize the main molecular biological events leading to the development of IR (Figure 2).

##### Endoplasmic Reticulum (ER) Stress

The main functions of the ER are synthesis, maturation, folding, and trafficking of secretory or membrane proteins. These are tightly regulated mechanisms and any disturbance of these processes can results in ER stress. The ER is especially sensitive to nutrient and energy homeostasis of the cell; thus, lifestyle factors such as diet or physical activity have significant impacts on the metabolism of this organelle [108]. Unresolved stress may lead to the disruption of the cellular and tissue homeostasis, which later contributes to the development of IR and metabolic syndrome (Figure 2A). ER stress triggers a compensatory signaling pathway, the unfolded protein response (UPR) [109].

UPR can be activated in three transmembrane transducers: PKR-like eukaryotic initiation factor 2α kinase (PERK), inositol requiring enzyme 1 (IRE-1), and activating transcription factor-6 (ATF6) (Figure 3).

The IRE-1, PERK and ATF6 membrane proteins are inactive while they are in complex with the ER stress sensor 78 kDa glucose-regulated protein (GRP78), better known as immunoglobulin heavy chain-binding protein (BiP) [109,110]. As a chaperone, BiP binds to misfolded and unfolded proteins in ER lumen. When the protein processing functions of ER fail, BiP is redirected to its client proteins, and is removed from the ER stress sensor. Thus, the three UPR transducers become active and trigger the three signaling branches of UPR (Figure 3).

On the IRE-1 pathway, activating phosphorylation of IRE-1α leads to the recruitment of tumor necrosis factor receptor-associated factor 2 (TRAF2), which then activates apoptosis signal-regulating kinase 1 (ASK1) and the stress-activated c-Jun N-terminal kinase (JNK) [111]. JNK then inactivates the insulin signaling by serine phosphorylation of IRS-1 (Figure 3), as well as inducing apoptosis, while inhibiting autophagy [103]. IRE-1α has endonuclease activity as well, which cleaves 26 base–pair sequences from the mRNA of X-box binding protein-1 (XBP1). This spliced variant sXBP-1 is facilitating the transcription of chaperons to enhance the folding capacity of ER. The misfolded proteins are removed from the ER, ubiquitinated, and transferred to proteasomal degradation; this process is called endoplasmic reticulum-associated degradation (ERAD). sXBP-1 also induces factors involved in ERAD to facilitate damaged protein clearance from ER lumen [109,112].The PERK pathway: after PERK activation by autophosphorylation, it subsequently phosphorylates the eukaryotic translation initiation factor 2α (eIF2α), generally inhibiting the translation of the proteins. However, some proteins will be translated properly in this unusual translational condition, such as the activating transcription factor 4 (ATF4), a member of cAMP-responsive element-binding protein (CREB) family [113], which induces the transcription of ER chaperons and the C/EBP homologous protein (CHOP), or, by another name, growth-arrest and DNA-damage-inducible protein (GADD153) (Figure 3) [23].In normal conditions, ATF6 occurs in its monomer, dimer, and oligomer forms in the ER membrane. Upon ER stress, it is reduced and forms monomers, which translocate to the Golgi apparatus, where it is activated by the cleavage of serine protease site-1 protease (S1P) and metalloprotease site 2 protease (S2P) [114]. Only the reduced monomer form can reach the Golgi and be released as an activated cytosolic fragment (ATF6f), which then enters the nucleus and enhances transcription of ER chaperons and CHOP (Figure 3) [110].

All three pathways of the UPR induced by ER stress aim to enhance the folding capacity of ER by inducing folding enzymes and chaperones, and lower the burden by reducing protein translation and promoting the degradation of misfolded proteins. If all these compensatory mechanisms do not restore ER homeostasis, apoptosis will be activated.

##### Inducing IR by Ser/Thr Phosphorylation of the IRS-1

IRS-1 comprises more than 70 serine/threonine (Ser/Thr) residues. The phosphorylation of IRS-1 on serine residues limit the phosphorylation of tyrosine residues, which are essential for binding to the regulatory subdomain and activation of PI3K. Thus, the downstream signaling pathway will be attenuated [115]. Increased Ser/Thr phosphorylation shifts IRS-1 to proteasomal degradation, and decreases IRS-1 levels with the result of a similarly attenuated insulin signaling [116,117].

ER stress induces IR through JNK activation on the IRE-1 pathway, and enhances apoptosis by phosphorylation on Ser307 residue of IRS-1, as was mentioned above [93,111]. JNK activation can be triggered by uncontrolled lipid accumulation in tissues through ER stress as well, which is called lipotoxicity [94,118,119]. However, both JNK activation and Ser307 phosphorylation of the IRS-1, in addition to the apoptosis in beta cells, can be attenuated by the anti-diabetic drug, metformin [93,120,121].

Inflammatory cytokines such as TNFα and IL-1β also can activate JNK and inhibitor of nuclear factor κB kinase β (IKKβ) [122], mediating inhibition of IRS-1 also by its Ser307 (hS312) phosphorylation (Figure 2B) [55,123]. TNFα as well as IL-6 inhibits autophosphorylation of IR on the tyrosine residues and promotes serine phosphorylation both in insulin receptor and IRS-1, leading to IR and glucose intolerance [124,125,126]. Thus, these cytokines mediate IR by blocking insulin action [127]. As mentioned earlier, elevated FFA levels can stimulate the production of TNFα and IL-6, resulting in the same outcome as described above [128].

Nutrient overload by the effect of a prolonged insulin/Rheb/mTORC1/S6K1 signaling induces Ser/Thr phosphorylation of IRS-1, as a negative feedback mechanism, resulting in IRS-1 degradation [55,129].

Atypical PKC λ/ζ activation by ceramides can also phosphorylate IRS-1 on Ser/Thr residues, contributing to IR [55,130].

The antibiotic anisomycin inhibits protein synthesis and can induce JNK activation, but not in an ER stress-dependent manner, which similarly, can lead to IRS-1 phosphorylation on Ser307 [103,131].

##### IR by Protein Tyrosine Phosphatase 1B (PTP1B)

PTP1B protein is attached to the ER via its C-terminal amino acid (AA) sequences, and it is a negative regulator of insulin and its signaling by dephosphorylating tyrosine residues of insulin receptor and IRS-1 [23,132]. Significant correlation was found between its coding gene polymorphisms and IR as well as T2DM. More than 20 single nucleotide polymorphisms (SNP) in the *PTP1B* gene was connected with elevated risk of type 2 diabetes [133]. Interestingly, while excessive production of ROS in mitochondria leads to ER stress and promotes IR, a certain level of ROS, generated during the normal metabolic processes, is required to normal insulin signaling [134]. PTPs are redox-sensitive enzymes, and ROS oxidize the crucial AA residues of PTPs, the process of which inactivates the enzyme leading to normal insulin signaling [23]. Supporting this, in *PTP1B* knockout mice, JNK activation and XBP-1 splicing were attenuated in the IRE-1 pathway [111,135]. Additionally, PTP1B attenuates the production of nitrogen monoxide (NO) as well and causes endothelial dysfunction in diabetes [136].

##### IR by Lipid Phosphatases—SHIP and PTEN

Lipid phosphatases are also negative regulators of insulin signaling by reducing PIP3 levels. The SH2 domain containing inositol 5′phosphatase 2 (SHIP2) hydrolyses PIP3 to PI(3,4)P2, while phosphatase and tensin homolog deleted on chromosome ten (PTEN) hydrolyses PIP3 to PI(4,5)P2, leading to attenuated insulin signal [23,137]. SHIP2 level is elevated in skeletal muscle and adipose tissue in insulin-resistant diabetic mice [137]. *SHIP2* gene polymorphisms, similarly to *PTP1B*, are associated with the presence of metabolic syndrome and type 2 diabetes [23]. *SHIP2* knockout mice are significantly more sensitive to insulin [138,139]. Similarly, in mice overexpressing *SHIP2*, the insulin-induced activation of Akt in the liver, adipose tissue, and skeletal muscle is attenuated [138,140].

PTEN was first identified as a tumor suppressor; its role in insulin signaling was discovered later. PTEN inhibits the conversion of PIP2 to PIP3, thus blocking the insulin action. *PTEN* gene polymorphisms are also associated with IR and T2DM [138,141]. Deletion of *PTEN* in the liver, skeletal muscle, fat, and pancreas enhanced the insulin action and was protective against to IR in a high-fat-diet model [23,142].

##### Role of Phosphoprotein Phosphatases in IR

Protein phosphatase 2A (PP2A) is a multimeric serine/threonine phosphatase with wide regulatory functions in cellular metabolisms as well as in the activity of protein kinase cascades. Kinases in main signaling pathways, such as MEK1, ERK1/2 and Akt, are inactivated by PP2A in the MAPK cascade as well as in insulin signaling [138,143]. Both protein phosphatase B (PP2B)—also known as calcineurin—and the PP2C family member PH domain and leucine-rich repeat protein phosphatases (PHLPPs) are inhibiting the insulin signal through the dephosphorylation of Akt [138,144].

##### Role of Adapter Proteins in IR

Adapter proteins play an important role as signal transducers and integrators in the insulin receptor pathway by bearing multiple modular protein binding domains. Several different adapter proteins were discovered to influence insulin resistance and lipid metabolism in the last couple of years [145].

Grb10 and Grb14 are adaptor proteins that decrease insulin receptor activity and reduce the access of substrates to activated receptors [138]. The loss of function phenotypes of Grb10 improves glucose tolerance, decrease adiposity, and increase muscle mass. The loss of function phenotypes of Grb14 improves insulin sensitivity and glucose tolerance [145].

Suppressors of cytokine signaling 1/3 (SOCS1/3) are E3 ubiquitine ligases target IRS proteins for degradation after their ubiquitination [145]. SOCS proteins competitively bind to the insulin receptor and link cytokine signaling to the IR [138].

Tribbles homolog 3 (Trb3) is a pseudokinase, which binds to Akt and inhibits its activation [138]. It interacts with a lot of transcription factors. It can be activated also through ER stress response on the IRE-1 pathway, and behaves as a feedback inhibitor of ATF4 [146].

Inositol phosphate IP7 inhibits Akt binding to plasma membrane, and reduces insulin signaling [138].

Cbl-b is a member of mammalian Cbl family proteins that acts as E3 ubiquitin ligase. It binds to the p85 regulatory subunit of PI3K in T cells [147]. It may cooperate with the IRS-1 as well, and can reduce insulin signaling [148].

Myeloid differentiation primary response gene 88 (MyD88) can be activated during inflammatory reactions, which often occur in obesity, and results in IR by the activation of IKKβ- nuclear factor κB (NFκB) pathway [149].

## 3. Molecular Targets in Insulin Signaling/Resistance in Preventive, Complementary, or Drug Treatment Strategies

### 3.1. Preventive or Complementary Treatment Options

Beside accessible drugs for T2DM or IR therapy, there are non-medical approaches as well, which either can prevent the development of these diseases or can be used as complementary therapy to increase treatment efficacy and/or quality of life of patients living with T2DM or IR. We summarize the possible therapeutic approaches, highlighting those targeted molecules in T2DM or IR-related signaling pathways, which are detailed above (Table 1).

#### 3.1.1. Diet and Supplementation

As we mentioned previously, inappropriately high calorie intake increases the amount of adipose tissue as well as assisting the accumulation of TGs in the liver and skeletal muscle (but in other tissues as well) in a higher amount than would be optimal for normal metabolic states of these organs and tissues. This change, if it affects the body for a long time, i.e., years, usually leads to MS, IR, and T2DM. In this section, we discuss in more detail the effects of some important components of nutrition to examine their impact on molecular changes observed in T2DM, but similarly in MS and IR, as these symptoms mostly appear together. We are not aiming for completeness, because many articles have been published in recent years, but would like to highlight the importance of this field.

Healthy, plant-based diets maximize the consumption of nutrient-dense plant food and minimize/decrease the amount of processed food with unnecessary oils and animal products. These are rich in fibre, phytochemicals, and micronutrients, and low in calories and saturated fat [150,151]. This nutritional composition has been found to be more effective within a few months in improving blood pressure, diabetes status, and lipid profile than metformin or exercise in monotherapy [152,153].

Phytochemicals, including polyphenols, carotenoids, isoprenoids, phytosterols, saponins, and polysaccharides, are produced by plant for their protection [154]. In different ways, polyphenols can inactivate NFκB, MAPK, and arachidonic acids pathways, therefore decreasing the level of inflammation and diminishing mitogenic pathways (Table 1, Figure 2A) [155].

**Table 1 ijms-25-09113-t001:** Diet and supplementation positively influences the development of T2DM/IR. T2DM: type 2 diabetes mellitus, IR: insulin resistance.

Diet/Supplementation	Physiological and Cellular Effect(s) on T2DM/IR	Cellular and Molecular Target(s)	Reference Number
**phytochemicals**			
polyphenols	•decreased inflammation and mitogenic processes	•inactivated NFκB, MAPK and arachidonic acids pathways	[155]

resveratrol	•diminish IR, decreased FFA, lipotoxycity and inflammation	•activates SIRT1 and AMPK; these activate PPARα and inhibit PPARγ, SREBP1c and NFκB, decrease TNFα and IL-6, management of AGEs	[19,20,104] [156]
	support gluconeogenesis/glucose uptake, ER stress- resistance, apoptosis	by activating FOXOs, increasing GLUT4, inhibiting mTORC1	

carotenoid			
astaxanthin	•attenuates IR, glucose intolerance, enhance exercise tolerance/FFA metabolism, stimulates anti-inflammatory effects	•oxidative phosphorylation and biogenesis through AMPK, through its antioxidant properties	[157]

lycopene	•promote angiogenesis for utilization of glucose and energy metabolism improves IR and vascular aging	•reactivation of SIRT1	[158]

phytosterols	•decrease the absorption of cholesterol, LDL cholesterol, promote insulin action through its receptors	•normalizing lipid metabolism increased GLUT4 translocation to plasma membrane	[159]

	inhibit mitochondrial dysfunction and ROS production	inhibit TNFα induced IKKβ/NFκB and JNK signaling	

phytosterols & saponins	•hypolipidemic and angioprotective effects		[159]
**micronutrients**			
vitamin D3	•increases glucose uptake inhibit oxidative stress	•inducing insulin-independent SIRT1/AMPK/IRS1/GLUT4 signaling pathway	[160] [161]

	optimal glucose and lipid homeostasis	adiponectin, AMPK	[20]

selenium	•necessary for normal inulin action, prevent IR	•support glucagon-like peptid receptor expression in beta cells necessary in the phosphorylation of insulin receptor	[162]

Zn&Mg	•necessary for normal inulin action, prevent IR	•necessary for autophosphorylation of insulin receptor, activation of PI3K and Akt and appropriate GLUT4 translocation to plasma membrane	[163,164]
**lipids**			
n-3 PUFAs	•increase β-oxidation of TGs and protein synthesis in skeletal muscle	•activates mTORC1 in skeletal muscle	[165]
	decrease lipogenesis in liver	inhibit diacylglycerol and SREBP1c	
	decrease lipogenesis in adipose tissue	activate AMPK, β-oxidation, increase adiponectin level	
**other natural compounds**	improve IR and support glucose utilization		[166]

Resveratrol is also a polyphenol known to activate sirtuin 1 (SIRT1), thereby AMPK as well, the molecules of which have the ability to diminish IR. They prevent the increase in FFA level and subsequent lipotoxicity as well as systemic inflammatory signals by activating peroxisome proliferator-activated receptor α (PPARα) and inhibiting PPARγ, SREBP1c, and NFκB, followed by a decrease in TNFα and interleukin-6 (IL-6) levels (Table 1, Figure 1B and Figure 2B). In addition to these, they foster gluconeogenesis as well as ER stress resistance and apoptosis through FOXOs, glucose uptake through increased GLUT4, but limit unnecessary protein synthesis by inhibiting mTORC1 (Table 1, Figure 1 and Figure 2A) [19,20,104]. Advanced glycated end-products (AGEs) are abundant in processed food as well as the consequence of high blood glucose level in patients with T2DM, inducing increased systemic inflammation in the human body [167]. Phenolic compounds appear to be effective in the managements of AGEs (Table 1) [156].

Astaxanthin, a carotenoid that is found in large amounts in marine organisms [168], significantly attenuated IR and glucose intolerance, enhanced exercise tolerance and exercise-induced FFA metabolism as well as enhanced mitochondrial oxidative phosphorylation and biogenesis through activation of AMPK pathways in muscle. Additionally, it stimulated anti-inflammatory effects in adipose tissue via its antioxidant properties (Table 1) [157]. The carotenoid lycopene found in tomato, red fruits, and vegetables could promote angiogenesis, utilization of glucose, and energy metabolism, thus, helping to improve IR in aging muscle and vascular aging via reactivation of SIRT1 (Table 1) [158].

Phytosterols are plant membrane sterols, similar in structure and action to cholesterol [169]. Cereals, vegetables, fruit, and nuts contain large amounts of phytosterols, an intake of which of around 2 g/day can decrease the absorption of cholesterol by 30–40% and of LDL cholesterol by around 10% [170]. Sterols are precursors of hormones in many organisms. The plant hormone β-sitosterol is a general component of drugs, normalizing lipid metabolism and reducing cholesterol levels. In combination with saponins, they have hypolipidemic and angioprotective activities (Table 1) [159]. Phytosterols inhibit TNFα-induced IKKβ/NFκB and JNK signaling as well as mitochondrial dysfunction and ROS production, while promoting insulin receptor signaling with GLUT4 translocation to plasma membrane (Table 1) [169,171].

Micronutients such as vitamins and so called trace elements, i.e., selenium, zinc, magnesium, cobalt, and copper, etc., are essential for enzymatic functions, formation of 3D structure of proteins, oxygen transport, and redox processes among others; thus, they are essential for adequate physiological function of the organisms [172,173].

Vitamin D has several roles in the human body, including Ca/P homeostasis, non-osteometabolic effects in glucose, lipid, and energy metabolism, and immune regulatory roles, as well as affecting epigenetic and transcription regulation through its vitamin D receptor [20,174]. Calcitriol, the hormonally active form of vitamin D, increases glucose uptake by inducing insulin-independent SIRT1/AMPK/IRS1/GLUT4 signaling pathway (Table 1) [160]. Moreover, it can inhibit oxidative stress through SIRT1/AMPK/GLUT4 signaling in high-glucose-treated adipocytes as well (Table 1) [161]. Vitamin D, through adiponectin and AMPK, supports optimal glucose and lipid homeostasis as well (Table 1) [20]. It is also important in the healthy regulation of the immune system, as was also observed during the COVID-19 pandemic [175,176,177].

Selenium is essential but at a very low level as a dietary intake, around 55–75 µg/day. Its biological action is primarily mediated by selenoproteins [178]. In regard to T2DM, the following should be mentioned: glutathione peroxidases, which remove hydroperoxides (e.g., hydrogen peroxide); thioredoxin reductases, which control redox functions; selenoprotein P (SELENOP), which transports Se to other tissues; and selenoprotein S (SELENOS), which is involved in ER function [179,180]. Epidemiology studies and meta-analysis found a significant positive but non-linear association between T2DM and serum Se [178] as well as Cu [181], which means both the low and high levels are connected to T2DM status. However, others concluded that under the condition of high extracellular glucose concentration and IR, the transcriptional activity of FOXO1 is dysregulated and may increase the biosynthesis of SELENOP, if sufficient Se is available, as well as gluconeogenetic enzymes. Thus, it is suggested that the high plasma level of Se, as a biomarker in T2DM, is more likely a consequence than a cause of diabetes [178]. Of note, selenium is important for the expression of the glucagon-like peptide (GLP) receptor in beta cells, which stimulates proinsulin gene expression in response to high glucose levels. Moreover, it is necessary in the regulation of phosphorylation of insulin receptor and its substrates in the insulin signal; thus, a deficiency may contribute to IR (Table 1) [162]. Nonetheless, Se supplementation beyond the adequate level is not recommended as an antioxidant treatment in T2DM [178]. Similarly to Se, both Zn and Mg deficiency contribute to IR, since in these states, the autophosphorylation of insulin receptor and activation of PI3K and Akt is compromised, coupled with blunted GLUT4 translocation to the plasma membrane, favoring hyperglycaemia (Table 1) [163,164].

Lipids and fatty acids (FA) are important components of nutrition. Although polyunsaturated fatty acids (PUFA) such as omega-3 (n-3) and omega-6 (n-6) PUFAs offer health benefits with their optimal n-6/n-3 ratio, their proportion can increase 4–10 times due to the Western diet, parallel with the level of saturated FA, which is detrimental to health [165]. Vegetable oils are more abundant in n-6 PUFA, while fish and fish oil are rich source of n-3 PUFAs, particularly eicosapentaenoic acid (EPA) and docosahexaenoic acid (DHA) [182]. N-3 PUFAs increase both the β-oxidation of TGs and protein synthesis by mTORC1 in skeletal muscle, but decrease lipogenesis by inhibiting diacylglycerol and SREBP1c in the liver, while in adipose tissue, they activate AMPK, PPARγ, and β-oxidation, and increase the level of adiponectin (Table 1) [165]. However, increased fat deposition and plasma FFA levels favor serine phosphorylation of IRS-1, leading to decreased translocation of GLUT4 to muscle plasma membranes [182].

Additionally, there are several natural products with potential hypoglycemic activity (Table 1) [166].

#### 3.1.2. Physical Activity and Exercise

There is well-known evidence for the preventive and therapeutic use of physical activity in several MS-related diseases [23,183,184,185]. Here, we detail those molecular aspects, which support the prevention and treatments of T2DM and IR, but all these may be useful in the management of other or related disease as well.

##### Systemic Effects of Physical Activity: Inflammation

The main effect of exercise is achieved through the reduction of systemic inflammation, body fat, and visceral adipose tissue, as well as accumulation of TG in skeletal muscle and other tissues, which are the main causes of MetS and related diseases [183,184]. Skeletal muscle, like adipose tissue, is an endocrine organ that secretes many factors, called myokines, which act on peripheral tissues, influencing their metabolic processes [23]. These myokines are members of the molecular family of cytokines, which are important in immune and inflammatory responses, regulation of cellular growth, and differentiation. This molecular group include ILs, chemokines (CCs, CXCs), TNF, interferons (IFNs), colony-stimulating factors (CSFs), and growth factors (GFs) [186].

Comparing the cytokine pattern of infection and exercise, the main difference can be found in the presence of TNFα and IL-1β as proinflammatory cytokines in early phase during infection but not in exercise, which is followed by an anti-inflammatory response with increased IL-6, IL-1ra, and IL-10 in both cases [184]. IL-6 is well-known for its pro-inflammatory effects secreted from adipose tissue. However, as was discovered, IL-6 has both pro- and anti-inflammatory properties depending on its signaling [187,188]. In classical IL-6 signal, IL-6 binds to its membrane-bound receptor, while in the one, which promotes disease pathogenesis, IL-6 binds to its soluble IL-6 receptor [189]. Earlier, it was thought that the increased level of IL-6 was caused by muscle damage during exercise, but later appeared that it is increased also in exercises with non-damaging muscle contractions [183]. It seems a single bout of intense exercise increases levels of TNFα, IL-1β, IL-8, macrophage inflammatory protein 1 (MIP-1), monocyte chemoattractant protein-1 (MCP-1), and IL-6, with secondary effects on IL-1ra and IL-10, while IFNγ decreases. On the contrary, a regular moderate exercise decreases the level of TNFα, IL-18, and IL-6, and increases the level of IL-1ra, IL-2, and IFNγ. The relative decrease in IL-6 in the plasma is probably because of the increased IL-6 receptor expression in skeletal muscle membrane [186]. IL-6 inhibits TNFα and IL-1β (thus limiting damage of the pancreas) as well as inducing the production of IL-1ra and IL-10, which further decrease the level of pro-inflammatory cytokines [183]. IL-6 increases the level of leptin, and activates AMPK both in adipocytes and skeletal muscle [190]. Moreover, exercise stimulates glucagon-like peptide-1 (GLP-1) secretion from intestinal L cells and pancreatic α-cells by IL-6, improving insulin secretion from β-cell [191].

##### Systemic Effects of Physical Activity: Hormone Secretion

Physical activity affects hormone secretion both through decreasing fat mass in the body as well as affecting endocrine organ function such as in the pancreas, adrenal gland, pituitary gland, and hypothalamus [23,183,192,193,194,195]. Adipose tissue secretes leptin and adiponectin among others [23]. Increased leptin level is connected with obesity, increased adiposity, and is parallel with leptin resistance, but exercise induces leptin sensitivity through increased IL-6 level as well [193]. Growth hormone (GH) in adults is responsible for optimal body composition and lipid metabolism; thus, deficiency of this hormone is related to disturbances in fat storage, mainly in visceral fat accumulation, bone mineral density as well as increased CVD risk [196]. Exercise stimulates cortisol via IL-6 [183], and induces prolactin and GH secretion as well [195].

##### Tissue-Specific Effects of Physical Activity and Exercise

Muscle contraction induces energy balance, activating both the SIRT1 and AMPK molecules with all their positive effects on energy/metabolic regulation [19,20]. Activated SIRT1 through AMPK and IRS-1 can upregulate GLUT4, resulting in increased glucose uptake in C2C12 myotubes [160]. Ex vivo study found the role of activated AMPK by exercise in GLUT4 transport [197], as well as in vivo studies confirming the role of exercise in the activation of AMPK/PGC-1α/GLUT4 in diabetes [198]. Peroxisome proliferator-activated receptor gamma co-activator 1α (PGC-1α) functions as a molecular switch in several cellular process, i.e., gluconeogenesis/glucose transport, glucogenolysis, FFA oxidation, muscle fibre-type switching, and oxidative phosphorylation, mitochondrial biogenesis/respiration in the liver, which are necessary to balance metabolic needs [199]. Moreover, as the causal relationship of PGC-1α dysregulation in skeletal muscle and energy homeostasis has successfully established with IR and T2DM, it has become a promising target of anti-diabetic therapy [92]. Nevertheless, exercise increases the amount of GLUT4 in both in the sarcolemma and transverse tubule membranes as well as on the cell surface, on the plasma membrane—while insulin increases GLUT4 level only in the latter —in the skeletal muscle of healthy humans and T2DM patients [200,201]. Along with these processes, both insulin and exercise decrease GLUT4 from subcellular compartments through general mobilization [70]. However, muscle contractions can deplete GLUT4 from transferrin receptor-positive endosomes as well, which are unaffected by insulin, supporting the idea that insulin and exercise activate different sets of recruitable transporter pools [202,203]. Exercise also can sensitize the muscle cell to insulin [204]. Exercise-induced SIRT1 through AMPK increases FFA oxidation in skeletal muscle, as well as in white adipose tissue and liver, mediating essential anti-diabetic function [205]. Exercise-induced adiponectin secretion activates AMPK and PPARα, of which, blunted actions lead to increased hepatic glucose production and hepatic IR [206]. In various tissues, AMPK can be activated by hormones, e.g., adiponectin, leptin, and pharmacological agents as metformin and thiazolidinediones [104].

### 3.2. Targets of Currently Used Drugs in Treatments of T2DM and IR

The biguanide metformin improve insulin sensitivity by activation of AMPK, however, an AMPK-independent mechanism can also be assumed (Table 2) [104,207,208,209].

Dipeptidyl-peptidase 4 (DPP-4) inhibitors enhance GLP-1 levels, thus, inducing insulin secretion, in addition to being involved in various biological processes (Table 2) [208,210].

The α-glucosidases inhibitors inhibit the breakdown of carbohydrates, thus attenuating glucose absorption in the small intestine and, subsequently, the postprandial blood glucose level [208].

GLP-1 receptor agonist liraglutid and semaglutid increase insulin sensitivity through weight loss (Table 2) [208].

It was recently recognized that glucocorticoids (GCs), a group of immunosuppressive drugs (i.e., dexamethasone, prednisolone, hydrocortisone), have beneficial roles in diabetes through their effects on peripheral IR and insulin secretion of β-cells. Function of GCs in the liver is partly mediated by PGC-1α (Table 2) [92].

Glucokinase (GK) is vital in many tissues regulating glucose metabolism and homeostasis. In pancreatic β-cells it mediates glucose-6-phosphate production and subsequent insulin release. In α-cells, it regulates amino acids and fatty acids, while in hepatocytes, GK has key role in glyconeogenesis, but is expressed in various tissues to influence glucose homeostasis directly or in an indirect manner [211].

Glimins have similarities in their structure with metformin. Imeglimin increases mitochondrial function, improves insulin secretion from pancreatic β-cells, reduces hepatic gluconeogenesis, and increases glucose uptake in skeletal muscle (Table 2) [212].

The sodium–glucose co-transporter-2 (SGLT2) inhibitor dapagliflozin effectively decreased serum TNFα and hepatic enzymes, and upregulated *SIRT1/PGC1α/FOXO1* liver gene expressions, and increased serum levels of AMPK, catalase, and SOD activity (Table 2) [208,213].

The sulfonylurea glimepiride stimulates β-cells in the pancreas to produce sufficient amounts of insulin [207]. Glycosylated sulfonylurea increased the expression of impaired IRS–PI3K–PKC–AKT–GLUT4 insulin signaling pathway genes (Table 2) [214].

Thiazolidinediones, similarly to metformin, have the ability to improve insulin sensitivity by activation of AMPK (Table 2) [104].

## 4. In Summary

T2DM and connected IR is increasing worldwide, and to turn back this process needs a more effective approach to prevent but also to treat these diseases. In our review, we aimed to summarize the normal insulin action as well as those pathways that mainly affect or are affected in the development of IR and T2DM. We detailed the beneficial molecular changes caused by some important components of nutrition and also by exercise. We presented that, in consensus with others, these are effective tools in daily practice as preventive as well as complementary therapies. These non-drug treatments have mostly the same targets in the human body, tissues, and organs as the developed drugs; however, these induce the entire network of regulatory mechanisms and proteins to restore unbalanced homeostasis. The main step for the success of this process is to rid the body of excess energy stored in fat tissues and absorbed from an inadequate amount or composition of food by nutritional changes and fat burning with adequate but regular exercise alongside necessary drug treatment if required. In the general population without clinical symptoms of IR and T2DM, adequate nutritional status and physical activity is suggested as prevention for maintaining health.

## Figures and Tables

**Figure 1 ijms-25-09113-f001:**
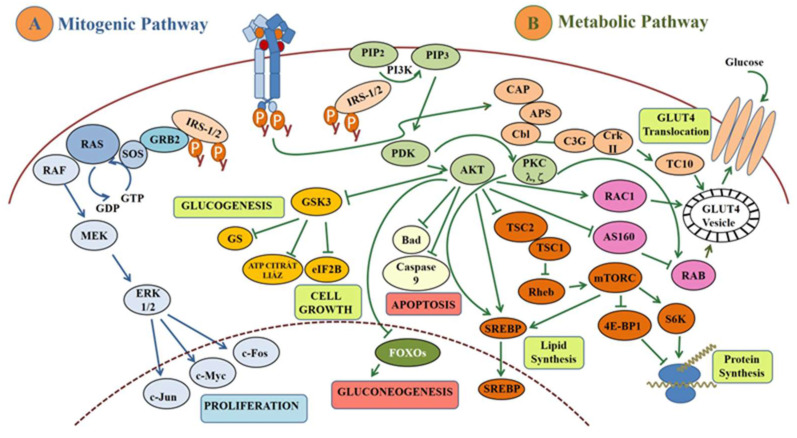
**Insulin signaling pathway**. The two main reactions to insulin in the cells are the mitogenic pathway (**A**) and metabolic pathway (**B**). Arrow (blue or green): activation, Truncated arrow (blue or green): inhibition, Blue color box/arrow: mitogenic pathway, Green arrow: metabolic pathway, Light green box: induced cellular process, Red box: inhibited cellular process. See text for detailed explanation. Abbreviations are listed at the end of the article.

**Figure 2 ijms-25-09113-f002:**
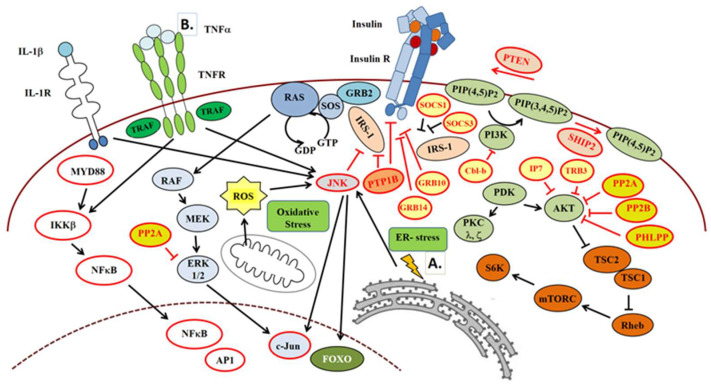
**Main pathways involved in the development of insulin resistance**. Intrinsic ((**A**)—ER stress) and extrinsic ((**B**)—tumor necrosis factor α (TNFα) receptor agonists) factors as well as inhibitory proteins involved in the progression of IR, among others, e.g., oxidative stress. Arrow: activation, Truncated arrow: inhibition. See further explanation in text.

**Figure 3 ijms-25-09113-f003:**
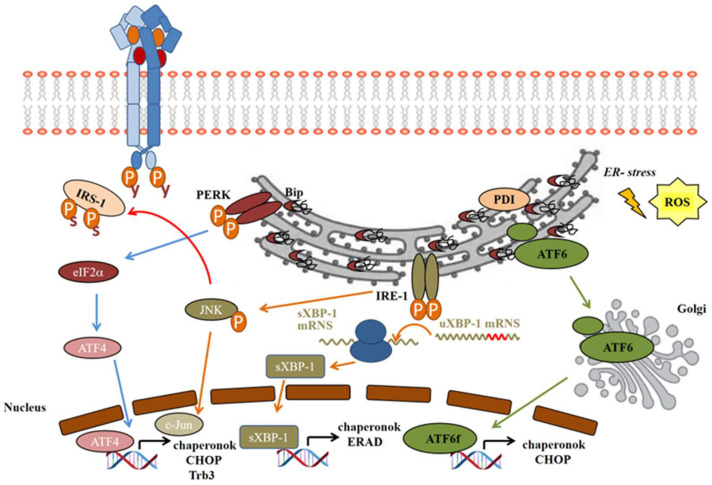
**The three main UPR pathways induced by ER stress.** Molecules and pathways involved in UPR, and their relation to IR signaling. Arrow (blue, orange and green): activation, Blue color arrow: PERK pathway, Orange color pathway: IRE-1 pathway, Green color arrow: ATF6 pathway.

**Table 2 ijms-25-09113-t002:** Drugs positively influence the development of T2DM/IR. T2DM: type 2 diabetes mellitus, IR: insulin resistance.

Drugs	Physiological and Cellular Effect(s) on T2DM/IR	Cellular and Molecular Target(s)	Reference Number
**Biguanides** Metformin	Improved IR, decreased inflammation	AMPK and AMPK-independent mechanisms	[104,207,208,209]
**DPP-4 inhibitors**	Increased insulin secretion	β-cells in pancreas	[208,210]
**GLP-1 receptor agonists** Liraglutid Semaglutid	Improved insulin production and weight loss	β-cells in pancreas	[191,208]
**Glucocorticoids**	Peripheral IR, insulin secretion	PGC-1α	[92]
**Glucokinase activators**	Glucose-6-phosphate production, subsequent insulin release	Glucose	[211]
**α-glucosidases inhibitors**	Inhibit breakdown of carbohydrates in small intestine	Alpha-glucosidase enzymes in small intestine	[208]
**Glimins** Imeglimin	Increased mitochondrial function, increased insulin secretion, reduced hepatic gluconeogenesis, increased glucose uptake in muscle	Mitochondria, ROS β-cells in pancreas AMPK	[212]
**SGLT2 inhibitor** Dapagliflozin	Decreased oxidative stress and inflammation, increased insulin sensitivity	SIRT1/AMPK/PGC1α/FOXO1 axis	[208,213]
**Sulfonylureas** Glimepiride Glycosylated sul fonylurea	Improved insulin production Improved insulin sensitivity	β-cells in pancreas IRS–PI3K–PKC–AKT–GLUT4	[207] [214]
**Thiazolidinediones**	Improved insulin resistance, decreased inflammation	AMPK	[104]

## Data Availability

Not applicable.

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
