# Peer review of "Molecular Aspects in the Development of Type 2 Diabetes and Possible Preventive and Complementary Therapies"

_ijms, 2024, doi:10.3390/ijms25169113_

Round 1

Reviewer 1 Report

Comments and Suggestions for Authors

This review entitled “Molecular aspects in the development of Type 2 diabetes and possible preventive and complementary therapies" by Simon-Szabó L. and colleagues was designed to highlight and summarize normal insulin action and the pathways that primarily influence the development of T2DM. The authors focused on two main topics (1) the introduction of insulin, insulin pathway, insulin resistance, and main pathways involved in the development of insulin resistance, as well as (2)molecular targets in insulin signaling/resistance in preventive, complementary or drug treatment strategies in this review. This study covers a wide range of molecular pathways (e.g. insulin, TNF-a, IL-1b, UPR signalling as well as ER stress) and will appeal to researchers engaged in basic and clinical research. Illustrations of signalling pathways make it easy for readers to comprehend. The summaries in the table also provide additional assistance to the reader in reaching conclusions about currently used drugs in the treatments of T2DM and IR.

Here are some minor suggestions,

(1)  Please summarize the findings in “3.1.1 Diet and supplementation” in a table.

(2)  IRS2 was not found in the figure 1.

(3)  The column "Effect(s) on T2DM/IR" and the column "Molecular Target(s)" in Table 1 are mixed up.

Author Response

Responses to Reviewer 1:

Thank you for your suggestions and comments on the manuscript ijms-3133949.

We corrected the manuscript based on your recommendations. The changes seen in the manuscript in blue texts for easier follow-up. Please see the detailed changes made in the manuscript provided below in blue text:

„Comments and Suggestions for Authors

This review entitled “Molecular aspects in the development of Type 2 diabetes and possible preventive and complementary therapies" by Simon-Szabó L. and colleagues was designed to highlight and summarize normal insulin action and the pathways that primarily influence the development of T2DM. The authors focused on two main topics (1) the introduction of insulin, insulin pathway, insulin resistance, and main pathways involved in the development of insulin resistance, as well as (2)molecular targets in insulin signaling/resistance in preventive, complementary or drug treatment strategies in this review. This study covers a wide range of molecular pathways (e.g. insulin, TNF-a, IL-1b, UPR signalling as well as ER stress) and will appeal to researchers engaged in basic and clinical research. Illustrations of signalling pathways make it easy for readers to comprehend. The summaries in the table also provide additional assistance to the reader in reaching conclusions about currently used drugs in the treatments of T2DM and IR.

Here are some minor suggestions,

(1) Please summarize the findings in “3.1.1 Diet and supplementation” in a table.

RE(1) We prepared and added an extra table summarizing the findings in 3.1.1 on page 11-14, and referring to this in the text by (Table 1).

(2) IRS2 was not found in the figure 1.

RE(2) We corrected the Figure 1 added to IRS2 there, labelling the molecule as IRS1/2 (page 4).

(3) The column "Effect(s) on T2DM/IR" and the column "Molecular Target(s)" in Table 1 are mixed up.

RE(3) In Table 2 (was earlier table 1) we complemented the 2nd- 3rd column titles - to be more specific and related to the content - to “Physiological and cellular effect(s) on T2DM/IR” in the 2nd column where the cellular and the physiological consequences of the drugs described, and in the 3rd to “Cellular and molecular target(s)” seen on page 17-18.

Reviewer 2 Report

Comments and Suggestions for Authors

Title: Molecular aspects in the development of Type 2 diabetes and possible preventive and complementary therapies

The main aim is to update and summarize molecular aspects that are affected by T2DM, IR, and their management through medication, nutrition and exercise, which act in the same molecular targets as the developed drugs, and can revert the damaged pathways.

The author has presented a quality study on T2DM, and its management through drugs and alternative routes.

Overall, the report has shown very few promising results and I wish that the author would consider making minor changes.

1.     Page 3-4: Kindly provide the references for T1DM is caused by autoimmune deterioration information.

2.     Kindly provide the details about IR. What’s the relationship of IR and T2DM? How does it lead to T2DM and why does it matter to discuss and study IR?

3.     Please provide more details and sequential flow on how IR leads to dyslipidaemia and finally to T2DM.

4.     Obesity data in Europe region - In what year did it find? Please mention it.

5.     Page 6: Please prepare a figure about inactive insulin (Pre-proinsulin) to active insulin and its binding to receptors, leading to activation of downstream signalling.

6.     Page 10: Elevated Glucose and FFA lead RE stress; kindly discuss and explain it in detail with recently published research articles.

7.     Table 1. Kindly add a section on the Pharmacological targets of drugs and medications and their side effects.

8.     Side effects presently available for T2DM management must be discussed in detail with relevant and recent published studies. After this, please explain why alternative therapies are needed to manage T2DM. 

Comments on the Quality of English Language

NA

Author Response

Responses to Reviewer 2:

Thank you for your suggestions and comments on the manuscript ijms-3133949.

We corrected the manuscript based on your recommendations. The changes seen in the manuscript in blue texts for easier follow-up. Please see the detailed changes made in the manuscript provided below in blue text:

„ Comments and Suggestions for Authors

Title: Molecular aspects in the development of Type 2 diabetes and possible preventive and complementary therapies

The main aim is to update and summarize molecular aspects that are affected by T2DM, IR, and their management through medication, nutrition and exercise, which act in the same molecular targets as the developed drugs, and can revert the damaged pathways.

The author has presented a quality study on T2DM, and its management through drugs and alternative routes.

Overall, the report has shown very few promising results and I wish that the author would consider making minor changes.

  1. Page 3-4: Kindly provide the references for T1DM is caused by autoimmune deterioration information. - We added references on page 2 for „T1DM caused by autoimmune deterioration” and to the next sentence as well (lines 71 and 72).
  2. Kindly provide the details about IR. What’s the relationship of IR and T2DM? How does it lead to T2DM and why does it matter to discuss and study IR? - We had already discussed IR in a separate section (in 2.4.3. from page 6 to 7) with explanations of the relationship between IR and T2DM as well as how it leads to T2DM. This section explains why it is important, and with detailed information it aims to foster targeted therapies. However, as we stated in line 265, molecular mechanisms of the development of IR is still not complete, which means it requires further investigation.
  3. Please provide more details and sequential flow on how IR leads to dyslipidaemia and finally to T2DM. - In the 2nd paragraph of section 2.4.3. we already described that with references (91,92) in line Y269-273, it is a widespread opinion the primary cause of the development of IR is the disturbance in the utilization of FFA as energy source. However, others suggest as primary role the defect of the proximal insulin pathway. In the 4th paragraph in this section in relation to IR-FFA/dyslipidaemia-T2DM we described in lines 286-290, 296-297, 298-302 and 306-507 briefly that, the increased glucose cannot taken-up properly, triggering increased insulin secretion and subsequent compensatory hyperinsulinaemia, which associated elevated level of FFA in the circulation. FFA triggers inflammation and all these cause b cell exhaustion, and apoptotic cell death with the end state where the missing insulin action is cannot compensated any more, which state is diagnosed as T2DM. Additionally, in section 2.2 we also described diabetic dyslipideamia, however its underlying pathophysiology is not fully understood (ref [34[). Also, in line 116-118 „Dyslipidaemia is a consequence of IR, resulted from the increased influx of enhanced level free fatty acid (FFA) to the liver due to glucose intolerance and compensatory elevated insulin levels [21,35,36].” However, we added these sentences to the end of the 2.2 section: “Although under physiological conditions insulin inhibits the secretion of triglyceride-rich VLDLs into the circulation, in the IR state, the increased influx of FFA into the liver increases the synthesis of Apo-B-containing hepatic TG and triglyceride-rich VLDLs. In addition, increased TG promotes the exchange of cholesterol to TG in HDL-C and LDL-C, converting them into small and dense particles, which alterations increases the clearance of HDL - the reverse transporter of cholesterol (from the periphery to liver) -, from the circulation, and the susceptibility of LDL to oxidation resulting its enhanced uptake by macrophages [36].”
  4. Obesity data in Europe region - In what year did it find? Please mention it. - We added the year 2022 in section 2.3. in lines 2-4 with necessary modification in the sentence.
  5. Page 6: Please prepare a figure about inactive insulin (Pre-proinsulin) to active insulin and its binding to receptors, leading to activation of downstream signalling. - The activation of insulin is out of scope of our review. In our review, we discuss the role of active insulin, which binds to its receptor. However, based on recent knowledge, each figure shows the 4-insulin-saturated T form of the insulin receptor, and two of the references [53] and [54] from 2020 and 2023 have been cited for those interested to follow the updated activation model.
  6. Page 10: Elevated Glucose and FFA lead RE stress; kindly discuss and explain it in detail with recently published research articles. - We described in line 290-292 that “increased amount of glucose and FFA cause gluco- and lipotoxicity and furthermore ER stress with the references where these processes are detailed. However, we inserted an additional sentence, after this one mentioned above, clarifying the details, “Increased cellular FFA during lipotoxicity disrupts ER-Golgi protein trafficking, causing protein overload in the ER, leading to ER stress and increases levels of ER stress markers, including the ER chaperone Bip and the apoptosis inducer CHOP, and increases the phosphorylation of IRS -1 at Ser307, which results in its increased degradation [93,101-103].”
  7. Table 1. Kindly add a section on the Pharmacological targets of drugs and medications and their side effects. - We have not added the pharmacological targets of drugs because of the molecular focus of this review instead of pharmacological and the references are there, if anyone is interested in more details and we also did not add their side effects. Side effects are long lists in the description of the drugs, which are not worth listing here, as they were not the main topic of our review from a molecular perspective and can be easily looked up by anyone. However, we have added a new reference [209] for metformin, as it is an older drug and other targets have recently been proposed in addition to its well-known target.
  8. Side effects presently available for T2DM management must be discussed in detail with relevant and recent published studies. After this, please explain why alternative therapies are needed to manage T2DM. - We already described why would be important complementary therapies in the “In summary” section in lanes 706-712: “We detailed the beneficial molecular changes caused by some important components of nutrition and also by exercise. We presented that, in consensus by others, these are effective tools in daily practice as preventive as well as complementary therapies. These non-drug treatments have mostly the same targets in the human body, tissues and organs, as the developed drugs, however, these induce the entire network of regulatory mechanisms and proteins to restore unbalanced homeostasis.”. Detailing all possible side effects are not relevant to the scope of this review. We aimed to detail molecular mechanisms, which can be good targets for new therapies and connect relevant complementary therapies, which can complement recent currently used drugs.
